# Resting-state alterations in behavioral variant frontotemporal dementia are related to the distribution of monoamine and GABA neurotransmitter systems

Lisa Hahn[1,2], Simon B Eickhoff[1,2], Karsten Mueller[3], Leonhard Schilbach[4,5], Henryk Barthel[6], Klaus Fassbender[7], Klaus Fliessbach[8,9], Johannes Kornhuber[10], Johannes Prudlo[9,11], Matthis Synofzik[9,12], Jens Wiltfang[9,13,14], Janine Diehl-Schmid[15,16], FTLD Consortium, Markus Otto[17,18], Juergen Dukart[1,2]*[†], Matthias L Schroeter[19,20][†]

[1]Institute of Neuroscience and Medicine, Brain & Behaviour (INM-7), Research Centre Jülich, Jülich, Germany; [2]Institute of Systems Neuroscience, Medical Faculty, Heinrich Heine University Düsseldorf, Düsseldorf, Germany; [3]Max Planck Institute for Human Cognitive and Brain Sciences, Leipzig, Germany; [4]LVR-Klinikum Düsseldorf, Düsseldorf, Germany; [5]Medical Faculty, Ludwig-Maximilians-Universität, München, Germany; [6]Department for Nuclear Medicine, University Hospital Leipzig, Leipzig, Germany; [7]Department of Neurology, Saarland University Hospital, Homburg, Germany; [8]Department of Psychiatry and Psychotherapy, University Hospital Bonn, Bonn, Germany; [9]German Center for Neurodegenerative Diseases (DZNE), Bonn, Germany; [10]Department of Psychiatry and Psychotherapy, University Hospital Erlangen, Friedrich-Alexander-University Erlangen-Nuremberg, Erlangen, Germany; [11]Department of Neurology, University Medicine Rostock, Rostock, Germany; [12]Department of Neurodegenerative Diseases, Center of Neurology, Hertie Institute for Clinical Brain Research, Tübingen, Germany; [13]Department of Psychiatry and Psychotherapy, University Medical Center Göttingen (UMG), Medical University Göttingen, Göttingen, Germany; [14]Neurosciences and Signaling Group, Institute of Biomedicine (iBiMED), Department of Medical Sciences, University of Aveiro, Aveiro, Portugal; [15]Department of Psychiatry and Psychotherapy, Technical University of Munich, Munich, Germany; [16]kbo-Inn-Salzach-Klinikum, Clinical Center for Psychiatry, Psychotherapy, Psychosomatic Medicine, Geriatrics and Neurology, Wasserburg/Inn, Germany; [17]Department of Neurology, Ulm University, Ulm, Germany; [18]Department of Neurology, Martin-Luther-University Halle-Wittenberg, Halle, Germany; [19]Department of Neurology, Max Planck Institute for Human Cognitive and Brain Sciences, Leipzig, Germany; [20]Clinic for Cognitive Neurology, University Hospital Leipzig, Leipzig, Germany

*For correspondence:
j.dukart@fz-juelich.de

[†]These authors contributed equally to this work

## Abstract

**Background:** Aside to clinical changes, behavioral variant frontotemporal dementia (bvFTD) is characterized by progressive structural and functional alterations in frontal and temporal regions. We examined if there is a selective vulnerability of specific neurotransmitter systems in bvFTD by evaluating the link between disease-related functional alterations and the spatial distribution of specific neurotransmitter systems and their underlying gene expression levels.

**Methods:** Maps of fractional amplitude of low-frequency fluctuations (fALFF) were derived as a measure of local activity from resting-state functional magnetic resonance imaging for 52 bvFTD patients (mean age = 61.5 ± 10.0 years; 14 females) and 22 healthy controls (HC) (mean age = 63.6 ± 11.9 years; 13 females). We tested if alterations of fALFF in patients co-localize with the non-pathological distribution of specific neurotransmitter systems and their coding mRNA gene expression. Furthermore, we evaluated if the strength of co-localization is associated with the observed clinical symptoms.

**Results:** Patients displayed significantly reduced fALFF in frontotemporal and frontoparietal regions. These alterations co-localized with the distribution of serotonin (5-HT1b and 5-HT2a) and γ-amino-butyric acid type A (GABAa) receptors, the norepinephrine transporter (NET), and their encoding mRNA gene expression. The strength of co-localization with NET was associated with cognitive symptoms and disease severity of bvFTD.

**Conclusions:** Local brain functional activity reductions in bvFTD followed the distribution of specific neurotransmitter systems indicating a selective vulnerability. These findings provide novel insight into the disease mechanisms underlying functional alterations. Our data-driven method opens the road to generate new hypotheses for pharmacological interventions in neurodegenerative diseases even beyond bvFTD.

**Funding:** This study has been supported by the German Consortium for Frontotemporal Lobar Degeneration, funded by the German Federal Ministry of Education and Research (BMBF; grant no. FKZ01GI1007A).

## Editor's evaluation

This study presents important findings linking structural and functional changes in frontotemporal dementia to underlying neurotransmitter systems. The evidence to support the claims is solid, however, relationships are relatively modest. This study will appeal to clinicians and neuroscientists who are interested in the potential effects of certain neurotransmitter systems on clinical features of frontotemporal dementia.

## Introduction

Frontotemporal lobar degeneration is the second most common type of early-onset dementia under the age of 65 years (*Harvey et al., 2003*). Its most common subtype, behavioral variant frontotemporal dementia (bvFTD), is characterized by detrimental changes in personality and behavior (*Pressman and Miller, 2014*). Patients can display both apathy and disinhibition, often combined with a lack of insight, and executive and socioemotional deficits (*Schroeter et al., 2011*; *Schroeter et al., 2012*). Despite striking and early symptoms, bvFTD patients are often (i.e. up to 50%) misdiagnosed as having a psychiatric illness rather than a neurodegenerative disease (*Woolley et al., 2011*).

In addition to the presence of symptoms, the diagnosis requires consideration of family history due to its frequent heritable component and examination of different neuroimaging modalities (*Pressman and Miller, 2014*; *Bang et al., 2015*; *Schroeter et al., 2014*; *Schroeter et al., 2008*). Whereas atrophy in frontoinsular areas only occurs in later disease stages, glucose hypometabolism in frontal, anterior cingulate, and anterior temporal regions visible with fluorodeoxyglucose positron emission tomography (FDG-PET) is already detectable from an early stage onwards (*Bang et al., 2015*; *Diehl-Schmid et al., 2007*). The fractional amplitude of low-frequency fluctuations (fALFF) is a resting-state functional magnetic resonance imaging (rsfMRI) derived measure with good test–retest reliability that closely correlates with FDG-PET (*Aiello et al., 2015*; *Holiga et al., 2018*; *Deng et al., 2022*). In frontotemporal dementia (FTD) patients, fALFF was reduced in inferior parietal, frontal lobes, and posterior cingulate cortex and holds great potential as MRI biomarker (*Premi et al., 2014*; *Borroni et al., 2018*). Low local fALFF activity in the left insula was linked to symptom deterioration (*Day et al., 2013*).

On a molecular level, frontotemporal lobar degeneration can be differentiated into three different subtypes based on abnormal protein deposition: tau (tau protein), transactive response DNA-binding protein with molecular weight 43 kDa (TDP-43), and FET (fused-in-sarcoma [FUS] and Ewing sarcoma [EWS] proteins, and TATA-binding protein-associated factor 15 [TAF15]) (*Bang et al., 2015*; *Haass and*

*Neumann, 2016*). Whereas tau and TDP pathologies each occur in half of the bvFTD patients, FUS pathology is very rare (*Whitwell et al., 2011*). Several possible mechanisms are discussed in the literature for the spread of these proteins throughout the brain, from a selective neuronal vulnerability (i.e. specific neurons being inherently more susceptible to the underlying disease-related mechanisms) to prion-like propagation of the respective proteins (*Walsh and Selkoe, 2016*; *Hock and Polymenidou, 2016*). The latter entails that misfolded proteins accumulate and induce a self-perpetuating process so that protein aggregates can spread and amplify, leading to gradual dysfunction and eventually death of neurons and glial cells (*Hock and Polymenidou, 2016*). For example, tau can cause presynaptic dysfunction prior to loss of function or cell death (*Zhou et al., 2017*), whereas overexpression of TDP-43 leads to impairment of presynaptic integrity (*Heyburn and Moussa, 2016*). The role of FET proteins is not fully understood, although their involvement in gene expression suggests a mechanism of altered RNA processing (*Svetoni et al., 2016*).

Neuronal connectivity plays a key role in the spread of pathology as it is thought to transmit along neural networks. Supporting the notion, previous studies also found an association between tau levels and functional connectivity in functionally connected brain regions, for example across normal aging and Alzheimer's disease (*Franzmeier et al., 2019*). Thereby, dopaminergic, serotonergic, glutamatergic, and GABAergic neurotransmission is affected. More specifically, current research indicates a deficit of neurons and receptors in these neurotransmitter systems (*Hock and Polymenidou, 2016*; *Huey et al., 2006*; *Murley and Rowe, 2018*). Furthermore, these deficits have been associated with clinical symptoms. For example, whereas GABAergic deficits have been associated with disinhibition, increased dopaminergic neurotransmission and altered serotonergic modulation of dopaminergic neurotransmission have been associated with agitated and aggressive behavior (*Engelborghs et al., 2008*; *Murley et al., 2020*). Another study related apathy to glucose hypometabolism in the ventral tegmental area, a hub of the dopaminergic network (*Schroeter et al., 2011*). Despite this compelling evidence of disease-related impairment at functional and molecular levels, the relationship between both remains poorly understood. It also remains unknown if the above neurotransmitter alterations reflect a disease-specific vulnerability of specific neuron populations or merely reflect a consequence of the ongoing neurodegeneration.

Based on the above findings, we hypothesize that the spatial distribution of fALFF and gray matter (GM) pathology in FTD will be related to the distribution of dopaminergic, serotonergic, and GABAergic neurotransmission. The aim of the current study was to gain novel insight into the disease mechanisms underlying functional and structural alterations in bvFTD by examining if there is a selective vulnerability of specific neurotransmitter systems. We evaluated the link between disease-related functional alterations and the spatial distribution of specific neurotransmitter systems and their underlying gene expression levels. In addition, we tested if these associations are linked to specific symptoms observed in this clinical population.

## Materials and methods
### Subjects

We included 52 Caucasian patients with bvFTD (mean age = 61.5 ± 10.0 years; 14 females) and 22 Caucasian age-matched healthy controls (HC) (mean age = 63.6 ± 11.9 years; 13 females) examined in nine centers of the German Consortium for Frontotemporal Lobar Degeneration (http://www.ftld.de; *Otto et al., 2011*) into this study. Details regarding the distribution of demographic characteristics across centers are reported in *Supplementary file 1a*. Diagnosis was based on established international diagnostic criteria (*Rascovsky et al., 2011*). Written informed consent was collected from each participant. The study was approved by the ethics committees of all universities involved in the German Consortium for Frontotemporal Lobar Degeneration (Ethics Committee University of Ulm approval number 20/10) and was in accordance with the latest version of the Declaration of Helsinki. The clinical and neuropsychological test data included the Mini Mental State Exam (MMSE), Verbal Fluency (VF; animals), Boston Naming Test (BNT), Trail Making Test B (TMT-B), Apathy Evaluation Scale (AES) (companion-rated) (*Glenn, 2005*), Frontal Systems Behavior Scale (FrSBe) (companion-rated) incl. subscales (executive function [EF], inhibition, and apathy) (*Grace and Malloy, 2001*), and Clinical Dementia Rating-Frontotemporal Lobar Degeneration scale-modified (CDR-FTLD) (*Knopman et al., 2008*). Demographic and neuropsychological test information for both groups is displayed in *Table 1*.

**Table 1.** Demographic and clinical information for bvFTD patients and HC.

| | bvFTD | | HC | | Group comparison | |
|---|---|---|---|---|---|---|
| Age (years) | 61.5 ± 10.0 | N = 52 | 63.6 ± 11.9 | N = 22 | t = −0.78 | p = 0.44 |
| Sex (male/female) | 38/14 | N = 52 | 9/13 | N = 22 | $X^2$ = 6.90 | p = 0.009* |
| Education (years) | 13.7 ± 3.19 | N = 50 | 13.5 ± 2.56 | N = 22 | t = 0.21 | p = 0.84 |
| Disease duration (years) | 3.98 ± 5.22 | N = 49 | – | – | – | – |
| Verbal Fluency (number of animals) | 12.2 ± 6.48 | N = 49 | 27.5 ± 4.77 | N = 19 | t = −9.30 | p < 0.001* |
| Boston Naming Test (total score) | 12.9 ± 2.79 | N = 49 | 15.0 ± 0.22 | N = 20 | t = −3.28 | p = 0.002* |
| Mini Mental State Exam (total score) | 25.2 ± 4.48 | N = 50 | 29.3 ± 0.64 | N = 20 | t = −4.03 | p < 0.001* |
| Trail Making Test B (s) | 179 ± 84.4 | N = 36 | 78.5 ± 22.0 | N = 19 | t = 5.09 | p < 0.001* |
| Apathy Evaluation Scale (total score) | 32.7 ± 11.0 | N = 35 | 9.50 ± 5.26 | N = 4 | t = 4.13 | p < 0.001* |
| Frontal Systems Behavior Scale (companion-rated, total frequency) | 72.7 ± 16.1 | N = 34 | 38.8 ± 12.3 | N = 5 | t = 4.49 | p < 0.001* |
| Frontal Systems Behavior Scale (companion-rated, total distress) | 66.9 ± 21.0 | N = 29 | 32 ± 9.56 | N = 4 | t = 3.25 | p = 0.003* |
| Frontal Systems Behavior Scale: Executive Function (companion-rated, total distress) | 23.6 ± 7.39 | N = 34 | 11.8 ± 4.50 | N = 4 | t = 3.11 | p = 0.004* |
| Clinical Dementia Rating-Frontotemporal Lobar Degeneration (total score) | 8.06 ± 3.92 | N = 45 | 0.05 ± 0.16 | N = 19 | t = 5.07 | p < 0.001* |

bvFTD – behavioral variant frontotemporal dementia, HC – healthy controls.
*Significant at p < 0.05.

## MRI acquisition and preprocessing of imaging data

Structural T1-weighted magnetization-prepared rapid gradient-echo MRI and rsfMRI (TR = 2000 ms, TE = 30 ms, FOV = 64 × 64 × 30, voxel size = 3 × 3 × 5 mm, 300 volumes) were acquired on 3T devices. *Table 2* reports center-specific imaging parameters confirming a high level of harmonization.

All initial preprocessing of imaging data was performed using SPM12 (*Penny et al., 2011*). To calculate voxel-wise GM volume (GMV), structural images were segmented, spatially normalized to MNI space, modulated, and smoothed by a Gaussian convolution kernel with 6 mm full-width at half maximum (FWHM). RsfMRI images were realigned, unwarped, co-registered to the structural image, spatially normalized to MNI space, and smoothed with a Gaussian convolution kernel with 6 mm FWHM. A GM mask was applied to reduce all analyses to GM tissue. Images were further processed in the REST toolbox (*Song et al., 2011*) version 1.8. Mean white matter and cerebrospinal fluid signals as wells as 24 motion parameters (Friston-24) were regressed out before computing voxel-based measures of interest. fALFF was calculated at each voxel as the root mean square of the blood oxygen level-dependent signal amplitude in the analysis frequency band (here: 0.01–0.08 Hz) divided by the amplitude in the entire frequency band (*Song et al., 2011*). fALFF is closely linked to FDG-PET and other measures of local metabolic activity as has been shown in healthy participants but also for example in Alzheimer's disease (*Deng et al., 2022*; *Marchitelli et al., 2018*).

## Contrast analyses of fALFF and GMV

To test for fALFF alterations, group comparisons were performed in SPM12 using a flexible-factorial design with group (bvFTD or HC) as a factor and age, sex, and site (i.e. one dummy variable per site) as covariates (*Huotari et al., 2019*). To test for group differences in GMV, the same design with addition of total intracranial volume (TIV) was used. Pairwise group t-contrasts (i.e. HC > bvFTD, bvFTD > HC) were evaluated for significance using an exact permutation-based cluster threshold (1000 permutations permuting group labels, p < 0.05) to control for multiple comparisons combined with an uncorrected voxel-threshold of p < 0.001. A permutation-based cluster threshold combined with an uncorrected voxel-threshold was used since standard correction methods such as a family wise error rate of 5% may lead to elevated false-positive rates (*Eklund et al., 2016*).

**Table 2.** Center-specific imaging parameters for structural and functional imaging.

| Center | rsfMRI | | | | | Structural MRI | | | |
|---|---|---|---|---|---|---|---|---|---|
| | TE (ms) | TR (ms) | FOV (X, Y, Z) | Voxel size (mm) | Volumes | TE (ms) | TR (ms) | FOV (X, Y, Z) | Voxel size (mm) |
| Bonn | 30 | 2000 | 64 × 64 × 30 | 3 × 3 × 5 | 300 | 3.06 | 2300 | 240 × 256 × 176 | 1 × 1 × 1 |
| Erlangen | 34 | 3000 | 64 × 64 × 30 | 3 × 3 × 5 | 300 | 2.98 | 2300 | 240 × 256 × 176 | 1 × 1 × 1 |
| Göttingen | 30 | 2000 | 64 × 64 × 30 | 3 × 3 × 6 | 300 | 2.96 | 2300 | 256 × 256 × 176 | 1 × 1 × 1 |
| Homburg | 30 | 2000 | 64 × 64 × 30 | 3 × 3 × 5 | 300 | 2.98 | 2300 | 240 × 256 × 176 | 1 × 1 × 1 |
| Leipzig | 30 | 2000 | 64 × 64 × 30 | 3 × 3 × 5 | 300 | 2.98 | 2300 | 240 × 256 × 176 | 1 × 1 × 1 |
| München (TU) | 30 | 2000 | 64 × 64 × 30 | 3 × 3 × 5 | 300 | 2.98 | 2300 | 240 × 256 × 176 | 1 × 1 × 1 |
| Rostock | 30 | 2200 | 64 × 64 × 34 | 3.5 × 3.5 × 3.5 | 300 | 4.82 | 2500 | 256 × 256 × 192 | 1 × 1 × 1 |
| Tübingen | 30 | 2000 | 64 × 64 × 30 | 3 × 3 × 5 | 300 | 2.96 | 2300 | 240 × 256 × 176 | 1 × 1 × 1 |
| Ulm | 30 | 2000 | 64 × 64 × 30 | 3 × 3 × 5 | 300 | 2.05 | 2300 | 240 × 256 × 192 | 1 × 1 × 1 |

rsfMRI – resting-state functional magnetic resonance imaging, MRI – magnetic resonance imaging, TE – echo time, TR – repetition time, FOV – field of view.

Bonn – University of Bonn, German Center for Neurodegenerative Diseases (DZNE), University Hospital Bonn.

Erlangen – University Hospital Erlangen.

Göttingen – Medical University Göttingen.

Homburg – Saarland University Hospital.

Leipzig – Max-Planck-Institute for Human Cognitive and Brain Sciences.

TU München – Technical University of Munich.

Rostock – University Hospital Rostock, German Center for Neurodegenerative Diseases (DZNE).

Tübingen – University Hospital Tübingen, Centre for Neurology, Hertie-Institute for Clinical Brain Research.

Ulm – Ulm University.

## Spatial correlation with neurotransmitter density maps

Confounding effects of age, sex, and site were regressed out from all images prior to further spatial correlation analyses. To test if fALFF alterations in bvFTD patients (relative to HC) are correlated with specific neurotransmitter systems, the JuSpace toolbox (*Dukart et al., 2021*) was used. The JuSpace toolbox allows for cross-modal spatial correlations of different neuroimaging modalities with nuclear imaging derived information about the relative density distribution of various neurotransmitter systems. All neurotransmitter maps were derived as averages from an independent healthy volunteer population and processed as described in the JuSpace publication including rescaling and normalization into the Montreal Neurological Institute space. More specifically, we wanted to test if the spatial structure of fALFF maps in patients relative to HC is similar to the distribution of nuclear imaging derived neurotransmitter maps from independent healthy volunteer populations included in the toolbox (5-HT1a receptor [*Savli et al., 2012*], 5-HT1b receptor [*Savli et al., 2012*], 5-HT2a receptor [*Savli et al., 2012*], serotonin transporter [5-HTT; *Savli et al., 2012*], D1 receptor [*Kaller et al., 2017*], D2 receptor [*Sandiego et al., 2015*], dopamine transporter [DAT; *Dukart et al., 2018*], Fluorodopa [FDOPA; *García Gómez et al., 2018*], γ-aminobutyric acid type A [GABAa] receptors [*Dukart et al., 2018*; *Myers et al., 2012*], μ-opioid [MU] receptors [*Aghourian et al., 2017*], and norepinephrine transporter [NET; *Hesse et al., 2017*]). Detailed information about the publicly available neurotransmitter maps is provided in *Supplementary file 1c*. In contrast to standard analyses of fMRI data, this analysis might provide novel insight into potential neurophysiological mechanisms underlying the observed correlations (*Dukart et al., 2021*). Using the toolbox, mean values were extracted from both neurotransmitter and fALFF maps using GM regions from the Neuromorphometrics atlas. Extracted mean regional values of the patients' fALFF maps were z-transformed relative to HC. Spearman correlation coefficients (Fisher's z-transformed) were calculated between these z-transformed fALFF maps of the patients and the spatial distribution of the respective neurotransmitter maps. Exact permutation-based p-values as implemented in JuSpace (10,000 permutations randomly assigning group labels using orthogonal permutations) were computed to test if the distribution of

the observed Fisher's z-transformed individual correlation coefficients significantly deviated from zero. Furthermore, adjustment for spatial autocorrelation was performed by computing partial correlation coefficients between fALFF and neurotransmitter maps adjusted for local GM probabilities estimated from the SPM12-provided TPM.nii (*Dukart et al., 2021*). All analyses were false discovery rate (FDR) corrected for the number of tests (i.e. the number of neurotransmitter maps). To further test if and how the observed fALFF co-localization patterns are explained by the underlying global atrophy, we repeated the co-localization analysis (p < 0.05) for the significant fALFF–neurotransmitter associations after controlling for total GMV. Additionally, the receiver operating characteristic (ROC) curves and corresponding areas under the curve (AUC) were calculated for patients (Fisher's z-transformed Spearman correlations) vs. HC (leave-one-out Z-score maps) to examine discriminability of the resulting fALFF–neurotransmitter correlations.

## Correlation with structural data

To test if the significant correlations observed between fALFF and neurotransmitter maps were driven by structural alterations (i.e. partial volume effects), the JuSpace analysis using the same parameters was repeated with local GMV incl. a correction for confounding effects of age, sex, site, and TIV. For further exploration, fALFF and GMV Fisher's z-transformed Spearman correlations as computed by the JuSpace toolbox were correlated with each other for each patient over all neurotransmitters. The median of those correlation coefficients was squared to calculate the variance in fALFF explained by GMV.

## Correlation with clinical data

To test if fALFF–neurotransmitter correlations are related to symptoms of bvFTD, we calculated Spearman correlation coefficients between significant fALFF–neurotransmitter correlations (Fisher's z-transformed Spearman correlation coefficients from JuSpace toolbox output) and clinical scales and neuropsychological test data (see *Table 1*). All analyses were FDR corrected for the number of tests. In addition, to test for the specificity of these associations we examined the direct associations between fALFF and the neuropsychological tests by computing Spearman correlations with the Eigenvariates extracted from the largest cluster of the HC > bvFTD SPM contrast.

## Association with gene expression profile maps

Furthermore, to test if fALFF alterations in bvFTD patients associated with specific neurotransmitter systems in the JuSpace analysis were also spatially correlated with their underlying mRNA gene expression profile maps, the MENGA toolbox (*Rizzo et al., 2016*; *Rizzo et al., 2014*) was used. Z-scores were calculated for the patients relative to HC using the confound-corrected images. The analyses were performed using 169 regions of interest and genes corresponding to each significantly associated neurotransmitter from the JuSpace analysis (5-HT1b: *HTR1B*; 5-HT2a: *HTR2A*; GABAa (19 subunits): *GABRA1–6*, *GABRB1–3*, *GABRG1–3*, *GABRR1–3*, *GABRD*, *GABRE*, *GABRP*, *GABRQ*; NET: *SLC6A2*). More specifically, Spearman correlation coefficients were calculated between the genomic values and re-sampled image values in the regions of interest for each patient and for each mRNA donor from the Allen Brain Atlas (*Hawrylycz et al., 2012*) separately. The Fisher's z-transformed correlation coefficients were averaged over the six mRNA donors. Bonferroni-corrected one-sample t-tests were performed for each neurotransmitter to examine, whether the correlation coefficient differed significantly from zero.

## Neurotransmitter-genomic correlations and gene differential stability

To further examine the association of fALFF–neurotransmitter correlations and mRNA gene expression profile maps, we explored the relationship between neurotransmitter maps included in the JuSpace toolbox and mRNA maps provided in the MENGA toolbox. The MENGA analysis was repeated using the same parameters to obtain Fisher's z-transformed Spearman correlation coefficients between the neurotransmitter maps and the mRNA gene expression profile maps.

To evaluate the robustness of the mRNA maps between donors, gene differential stability was estimated by computing the Fisher's z-transformed Spearman correlation coefficients between the genomic values of each of the six mRNA donors, which were then averaged (*Hawrylycz et al., 2012*).

# Results

## Contrast analysis of fALFF and GMV

First, we tested for group differences in fALFF between HC and patients. Compared to HC, bvFTD patients showed a significantly reduced fALFF signal in frontoparietal and frontotemporal regions (*Figure 1A*). Furthermore, patients also showed reduced GMV in medial and lateral prefrontal, insular, temporal, anterior caudate, and thalamic regions in comparison to HC (*Figure 1B*). For a detailed representation of the thresholded fALFF and GMV t-maps, see *Figure 1—figure supplement 1*. Cluster size, peak-level MNI coordinates, and corresponding anatomical regions incl. the additional fALFF analysis with correction for total GMV are reported in *Supplementary file 1d*. For the distribution of Eigenvariates for the two groups in both modalities, see *Figure 1—figure supplement 2*.

## Spatial correlation with neurotransmitter maps

We performed correlation analyses to test if fALFF alterations in bvFTD significantly co-localize with the spatial distribution of specific neurotransmitter systems. fALFF alterations in bvFTD as compared to HC were significantly associated with the spatial distribution of 5-HT1b (mean $r = -0.21$, p < 0.001), 5-HT2a (mean $r = -0.16$, p = 0.0014), GABAa (mean $r = -0.12$, p = 0.0149), and NET (mean $r = -0.13$, p = 0.0157) ($p_{FDR} = 0.0157$; *Figure 2A*). The directionality of these findings (i.e. a negative correlation) suggest bvFTD displayed stronger reductions in fALFF relative to HC in areas which are associated with a higher non-pathological density of respective receptors and transporters. When controlling for total GMV, the co-localization findings remained significant except for the co-localization with GABAa. The AUC resulting from the ROC curves between Spearman correlation coefficients of patients and controls revealed a good discrimination for 5-HT1b (AUC = 0.74) and 5-HT2a (AUC = 0.71) and a fair discrimination for GABAa (AUC = 0.68) and NET (AUC = 0.67) (*Figure 3A*).

Next, we tested if similar co-localization patterns are observed with GMV. GMV alterations in bvFTD were not significantly associated with any of the neurotransmitter systems (*Figure 2B*). fALFF–neurotransmitter and GMV–neurotransmitter correlations displayed a positive yet weak association with structural alterations explaining only 10% of variance in the fALFF alterations (*Figure 3B*). All correlations and their corresponding permutation-based p-values incl. the analysis utilizing fALFF images additionally corrected for total GMV are provided in *Supplementary file 1c*. To exclude a potential bias caused by the collection of imaging data at different sites, we performed a Kruskal–Wallis test to examine differences on the Fisher's z-transformed correlations coefficients across sites. No significant differences ($X^2 = 6.34$, p = 0.50, df = 7) were found among the sites.

## Relationship to clinical symptoms

Furthermore, we tested if the significant fALFF–neurotransmitter correlation coefficients are also associated with symptoms or test results of bvFTD. After FDR correction (p = 0.0085), the strength of fALFF co-localization with NET distribution was significantly associated with VF (mean $r = 0.37$, p = 0.0086; N = 49; *Figure 2C*) and MMSE (mean $r = 0.40$, p = 0.0039; N = 50; *Figure 2D*). The positive correlation coefficients suggest that more negative correlations between fALFF and neurotransmitter maps were associated with lower test performance, that is the higher/more fALFF reductions in areas with high neurotransmitter density, the lower the test performance. Associations with other neuropsychological tests were not significant (*Supplementary file 1c*). We also tested if Eigenvariates extracted from the largest cluster of the HC > bvFTD contrast correlated with the specific symptoms of bvFTD (*Supplementary file 1f*). None of the correlations remained significant after correction for multiple comparisons.

## Association with gene expression profile maps

Next, we evaluated if co-localization of fALFF is also observed with mRNA gene expression underlying the significantly associated neurotransmitter systems. For genes encoding the 19 GABAa subunits, we first evaluated the variability between the subunits regarding their fALFF–mRNA correlations, their correlation with GABAa density and their mRNA autocorrelations (see *Figure 2—figure supplement 1* and *Figure 3—figure supplement 1*). As the variability between the genes was high, we limited the analyses to genes encoding the three main subunits (GABRA1, GABRB1, and GABRG1).

Correlations of fALFF alterations with mRNA gene expression profile maps in bvFTD relative to HC differed significantly from zero for *HTR1B* (encoding the 5-HT1b receptor; mean $r = -0.02$, p

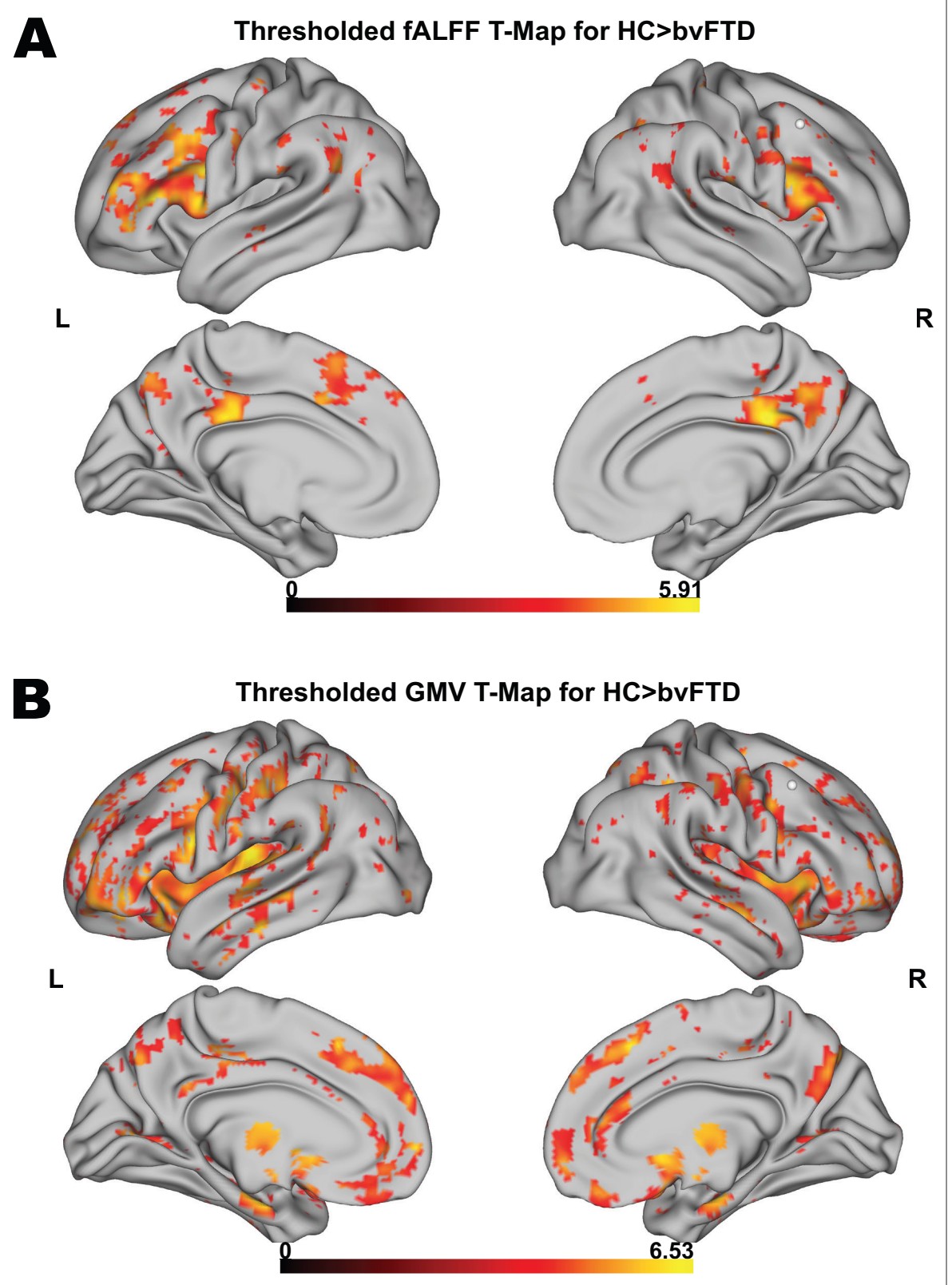

**Figure 1.** Voxel-wise results for fractional amplitude of low-frequency fluctuation (fALFF) and gray matter volume (GMV) group comparisons. Thresholded fALFF t-map (**A**) and thresholded GMV t-map (**B**) for healthy control (HC; N = 22) > behavioral variant frontotemporal dementia (bvFTD; N = 52) using a permutation-based threshold (1000 permutations permuting group labels) at cluster-level p < 0.05 and voxel-level p < 0.001.

*Figure 1 continued on next page*

*Figure 1 continued*

The online version of this article includes the following source data and figure supplement(s) for figure 1:

**Figure supplement 1.** Detailed voxel-wise results for fractional amplitude of low-frequency fluctuation (fALFF) and gray matter volume (GMV) group comparisons.

**Figure supplement 2.** Eigenvariates from fractional amplitude of low-frequency fluctuation (fALFF) and gray matter volume (GMV) for behavioral variant frontotemporal dementia (bvFTD) patients and controls.

**Figure supplement 2—source data 1.** Eigenvariates of fractional amplitude of low-frequency fluctuation (fALFF) and gray matter volume (GMV) for largest clusters of healthy control (HC) > behavioral variant frontotemporal dementia (bvFTD) *t*-contrasts shown in *Figure 1—figure supplement 2*.

= 0.0144), *HTR2A* (encoding the 5-HT2a receptor; mean $r = -0.04$, $p < 0.001$), *GABRB1* (encoding subunit of the GABAa receptor; mean $r = -0.08$, $p < 0.001$) and *SLC6A2* (encoding NET; mean $r = 0.06$, $p < 0.001$), but not for GABRA1 (encoding subunit of the GABAa receptor; mean $r = 0.02$, $p = 0.1414$) and GABRG1 (encoding subunit of the GABAa receptor; mean $r = -0.03$, $p = 0.0730$) (*Figure 2G*). Thereby, correlations were negative for *HTR1B*, *HTR2A*, and *GABRB1*, that is fALFF was reduced in areas with higher expression of respective genes, and positive for *SLC6A2*.

Furthermore, we tested if there was an association between the neurotransmitter maps included in the JuSpace toolbox and the mRNA gene expression profile maps provided in the MENGA toolbox that were both derived from independent healthy volunteer populations. The correlations between spatial distributions of 5-HT1b, 5-HT2a, GABAa, and NET, and corresponding mRNA gene expression profile maps were positive (5-HT1b/*HTR1B*: mean $r = 0.12$; 5-HT2a/*HTR2A*: mean $r = 0.20$; GABAa/*GABRA1*: mean $r = 0.14$; GABAa/*GABRB1*: mean $r = 0.14$; NET/*SLC6A2*: mean $r = 0.02$) with exception of the GABRG1 gene (GABAa/*GABRG1*: mean $r = -0.13$) (*Figure 3C*). Positive correlation coefficients suggest that higher neurotransmitter density was associated with higher expression of those neurotransmitters.

Lastly, to evaluate the robustness of the mRNA analyses (i.e. gene differential stability), genomic autocorrelations were calculated. The genomic autocorrelation was high for *GABRB1* (mean $r = 0.92$) and *GABRG1* (mean $r = 0.64$), small for *HTR1B* (mean $r = 0.23$), *SLC6A2* (mean $r = 0.22$), and *GABRA1* (mean $r = 0.21$), and very small for *HTR2A* (mean $r = 0.05$) (*Figure 3D*).

## Discussion

In the current study, we examined if there is a selective vulnerability of specific neurotransmitter systems in bvFTD to gain novel insight into the disease mechanisms underlying functional and structural alterations. More specifically, we evaluated if fALFF alterations in bvFTD co-localize with specific neurotransmitter systems. We found a significant spatial co-localization between fALFF alterations in patients and the in vivo derived distribution of specific receptors and transporters covering serotonergic, norepinephrinergic, and GABAergic neurotransmission. These fALFF–neurotransmitter associations were also observed at the mRNA expression level and their strength correlated with specific clinical symptoms. All of the observed co-localizations with in vivo derived neurotransmitter estimates were negative with lower fALFF values in bvFTD being associated with a higher density of the respective receptors and transporters in health. The directionality of these findings supports the notion of higher vulnerability of respective networks to disease-related alterations. These findings are also largely in line with previous research concerning FTD showing alterations in all of the respective neurotransmitter systems (*Huey et al., 2006*; *Murley and Rowe, 2018*).

The in vivo co-localization findings might also support the notion that propagation of proteins involved in bvFTD may align with specific neurotransmitter systems (*Hock and Polymenidou, 2016*). With regard to other brain disorders, linking functional connectivity with receptor density and expression, recent studies found an association between functional connectivity and receptor availability in schizophrenia, and an association between structural–functional decoupling and receptor gene expression in Parkinson's disease (*Zarkali et al., 2021*; *Horga et al., 2016*). A potential mechanism for the selective vulnerability of specific neurotransmitter systems is the propagation of proteins along functionally connected networks that has been previously demonstrated for various neurodegenerative diseases (*Zhou et al., 2012*; *Seeley et al., 2009*). For example, in Alzheimer's disease and normal aging, tau levels closely correlated with functional connectivity (*Franzmeier et al., 2019*). We found moderate to large AUC when using the strength of the identified co-localizations for differentiation

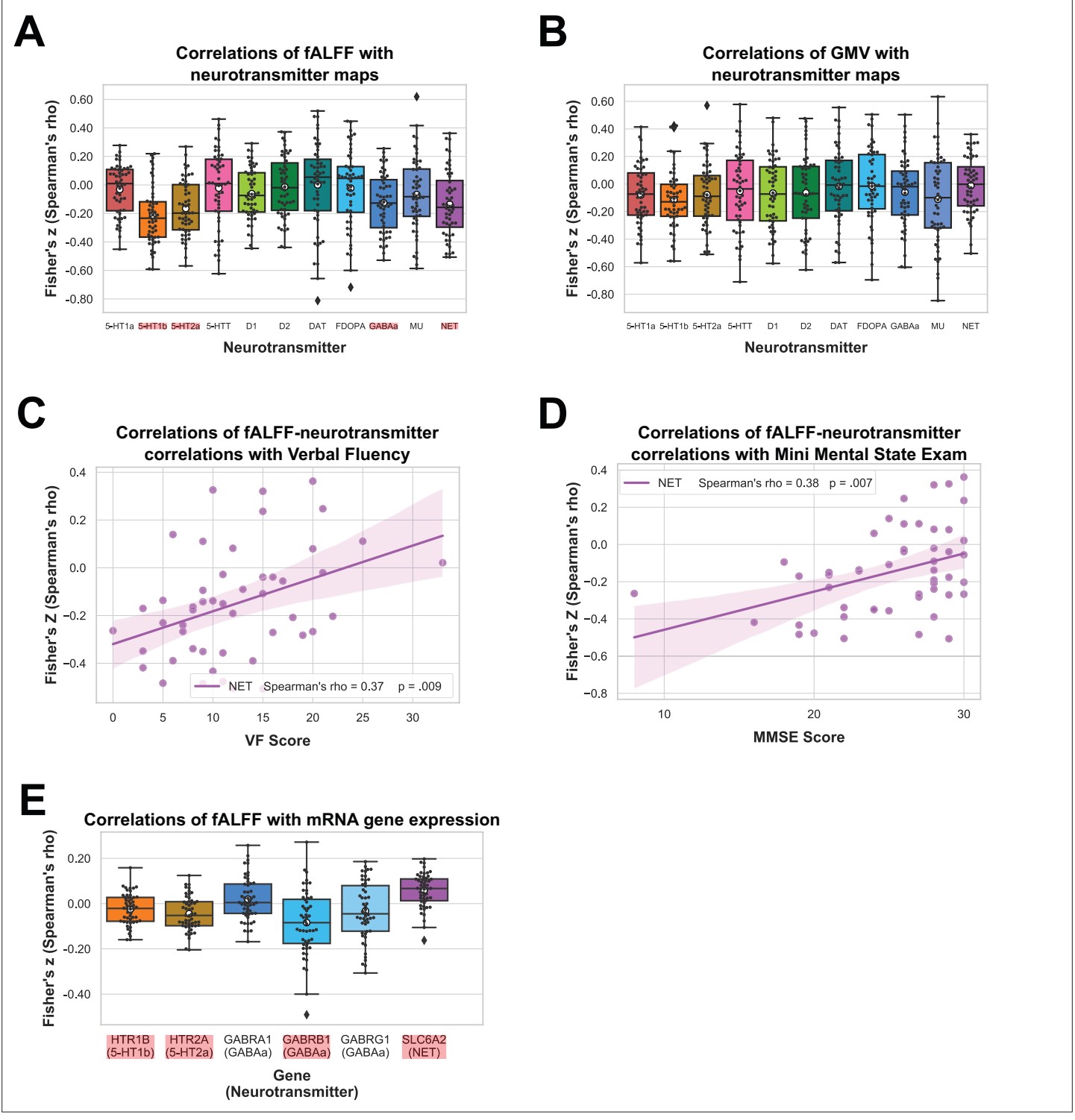

**Figure 2.** Results of spatial correlation analyses with in vivo and mRNA data. Correlation of fractional amplitude of low-frequency fluctuation (fALFF) (**A**) and gray matter volume (GMV) (**B**) with spatial distribution of neurotransmitter systems incl. 95% confidence intervals. Correlations of Verbal Fluency (*N* = 49) (**C**) and Mini Mental State Exam (*N* = 50) (**D**) with fALFF–neurotransmitter strength of association incl. bootstrapped 95% confidence intervals. Correlations of fALFF with mRNA gene expression maps (*N* = 52) (**E**). Statistically significant correlations in (**A**), (**B**), and (**E**) are marked in red and means are represented by white circles. Black circles in (**A**), (**B**), and (**E**) represent individual Fisher's z-transformed Spearman correlation coefficients for each patient (*N* = 52) relative to controls with each neurotransmitter map. Colored circles in (**C**) and (**D**) represent individual Fisher's z-transformed Spearman correlation coefficients between fALFF–neurotransmitter correlations and each neuropsychological scale. The statistical significance of all correlation coefficients was evaluated at p < 0.05 including FDR correction for (**A**), (**B**), and (**E**).

*Figure 2 continued on next page*

*Figure 2 continued*

The online version of this article includes the following source data and figure supplement(s) for figure 2:

**Source data 1.** Fisher's *z*-transformed Spearman correlation coefficients shown in *Figure 2A–E*.

**Figure supplement 1.** Results of spatial correlation of fractional amplitude of low-frequency fluctuation (fALFF) with mRNA gene expression maps of all γ-aminobutyric acid type A (GABAa) subunits.

**Figure supplement 1—source data 1.** Fisher's *z*-transformed Spearman correlation coefficients of fractional amplitude of low-frequency fluctuation (fALFF) with mRNA gene expression of all γ-aminobutyric acid type A (GABAa) subunits shown in *Figure 2—figure supplement 1*.

between patients and HC suggesting that these findings may represent a measure of the affectedness of respective neurotransmitter systems. In bvFTD, neurodegeneration is thought to progress through the salience network involved in socioemotional tasks, which comprises the anterior cingulate and frontoinsular cortex, as well as the amygdala and the striatum (*Bang et al., 2015*; *Hock and Polymenidou, 2016*). The three neurotransmitter systems found to be deficient in our sample are relevant for the functioning of these structures (anterior cingulate cortex: e.g. serotonin and norepinephrine, *Tian et al., 2017*; *Koga et al., 2020*; amygdala: e.g. GABA and serotonin, *Castro-Sierra et al., 2005*; striatum: e.g. GABA, *Semba et al., 1987*). Although spread of misfolded proteins through the salience network provides a potential disease mechanism, further research of the exact mechanisms involved is needed.

For GMV, we did not find any significant co-localization with specific neurotransmitter systems. As the correlations with GMV showed a distinct pattern to fALFF and the variance explained by GMV in the observed fALFF–neurotransmitter associations was small, the observed associations with fALFF seem to be driven indeed by functional alterations and not by the underlying atrophy of respective regions. As propagation of misfolded proteins leads to a gradual dysfunction and eventually cell death (*Hock and Polymenidou, 2016*), some regions displaying high density of a specific neurotransmitter might suffer dysfunction (i.e. functional alterations), whereas others might already be exposed to cell death (i.e. structural alterations/atrophy). An interesting future direction might compose integration of structural connectivity as measured by diffusion tensor imaging. A study by *Dopper et al., 2014* showed reduced fractional anisotropy in healthy individuals carrying mutations compared to non-carriers (*Dopper et al., 2014*). Given that there were structural connectivity differences even before disease onset, it would be of interest to re-examine structural connectivity differences between HC and patients (i.e. after disease onset). Repeating the neurotransmitter analyses might facilitate understanding of the underlying disease mechanism.

The strength of co-localization of fALFF with NET was correlated with VF and MMSE, both being impaired in patients with bvFTD (*Schroeter et al., 2012*; *Diehl and Kurz, 2002*; *Schroeter et al., 2018*). Thereby, a stronger negative co-localization (i.e. lower fALFF in patients in high-density regions in health) was moderately associated with decreased test performance. Similarly, a correlation between MMSE and NE plasma concentration has been previously reported in Alzheimer's disease (*Pillet et al., 2020*). Combined, these findings point to a potentially more general role of norepinephrinergic neurotransmission in cognitive decline observed across different dementia syndromes. This interpretation is in line with the recently proposed role of the locus coeruleus, the source of norepinephrine in the brain, in regulating processes of learning, memory, and attention (*Tsukahara and Engle, 2021*). In contrast to the study by *Murley et al., 2020* who reported an association of GABA concentrations in the inferior frontal gyrus in FTD with disinhibition, we did not find this association. Beside the use of different methodology, a potential explanation may constitute the use of different inhibition measures. Whereas we measured disinhibition using the FrSBe, *Murley et al., 2020* used a stop-signal task.

Although, except for α1 and γ1 GABAa subunits, all of the co-localizations with fALFF identified with in vivo estimates were also significant at the respective mRNA gene expression level, we found correlation coefficients of both directionalities. Interestingly, whereas these correlations were solely negative for the in vivo derived maps, the correlations with gene expression profile maps were positive for NET, and negative for 5-HT1b, 5-HT2a, and β1 GABAa subunit. Thus, for NET, we observed higher fALFF values in bvFTD patients in areas with high mRNA gene expression in health, whereas for 5-HT1b, 5-HT2a, and β1 GABAa subunit we observed lower fALFF values in bvFTD patients in areas with high mRNA gene expression in health. One explanation for these seemingly contradictory

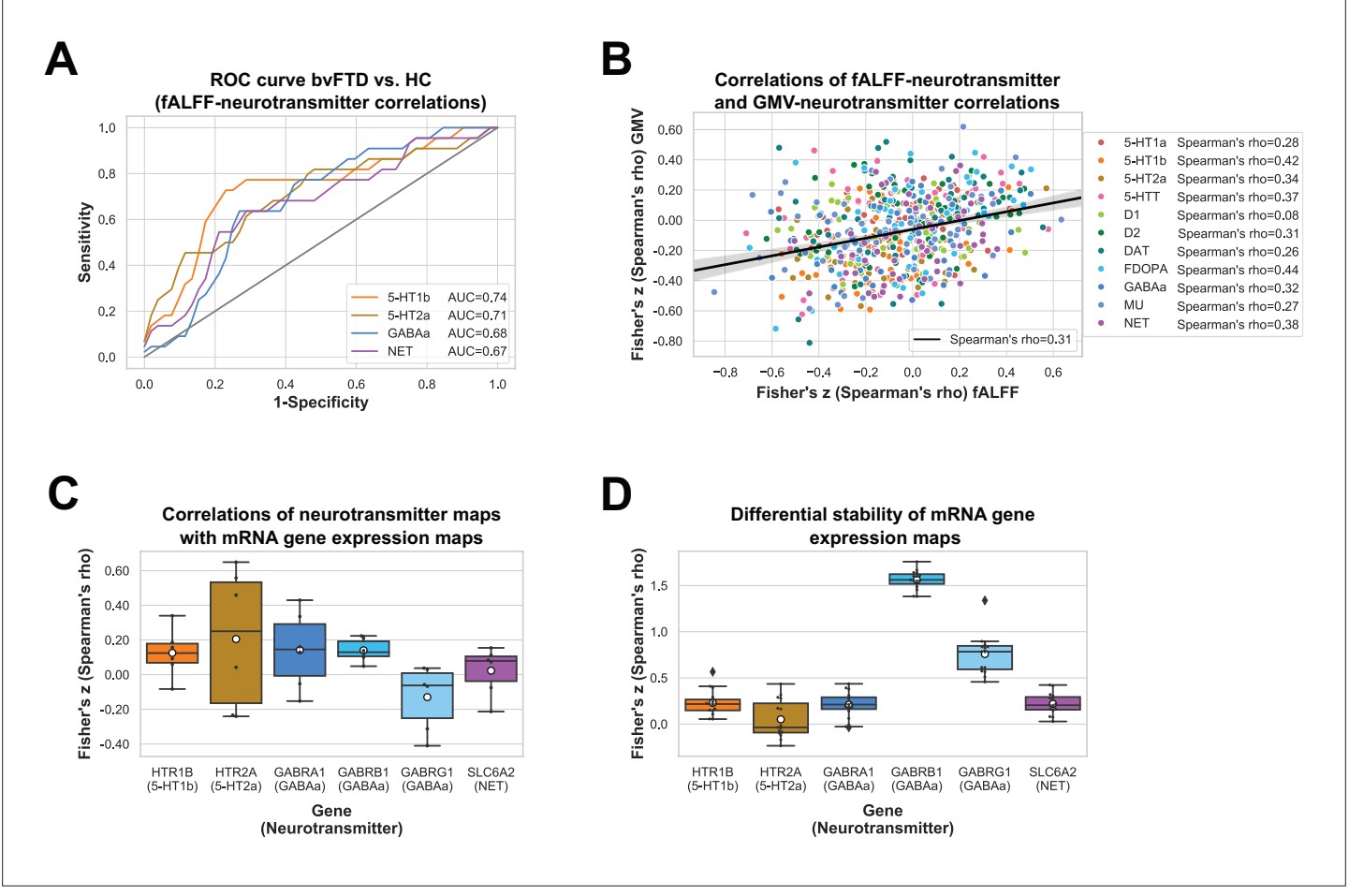

**Figure 3.** Results for fractional amplitude of low-frequency fluctuation (fALFF)–neurotransmitter receiver operating characteristic (ROC) curve, correlations of fALFF–neurotransmitter and gray matter volume (GMV)–neurotransmitter correlations, correlations of neurotransmitter and mRNA gene expression maps, and autocorrelations of mRNA gene expression maps. ROC curves for healthy controls (HC) vs. behavioral variant frontotemporal dementia (bvFTD) patients are displayed for significant fALFF–neurotransmitter correlations ($N_{bvFTD}$ = 52, $N_{HC}$ = 22) (**A**). Spearman correlation coefficients of fALFF–neurotransmitter and GMV–neurotransmitter correlations are displayed for each patient and each significant neurotransmitter ($N$ = 52) (**B**). Spearman correlation coefficients of neurotransmitter and mRNA gene expression maps (**C**) and autocorrelations of mRNA gene expression maps averaged across mRNA donors ($N$ = 6) (**D**) are displayed for significant fALFF–neurotransmitter associations incl. 95% confidence intervals.

The online version of this article includes the following source data and figure supplement(s) for figure 3:

**Source data 1.** Sensitivity and 1 − specificity shown in **Figure 3A**, fractional amplitude of low-frequency fluctuation (fALFF)–neurotransmitter and gray matter volume (GMV)–neurotransmitter Fisher's $z$-transformed Spearman correlation coefficients shown in **Figure 3B**, and Fisher's $z$-transformed Spearman correlation coefficients of neurotransmitter and mRNA gene expression maps shown in **Figure 3C ,D**.

**Figure supplement 1—source data 1.** Fisher's $z$-transformed Spearman correlation coefficients of neurotransmitter and mRNA gene expression maps for all γ-aminobutyric acid type A (GABAa) subunits shown in **Figure 3—figure supplement 1**.

**Figure supplement 1.** Results for correlations of neurotransmitter and mRNA gene expression mapsof all γ-aminobutyric acid type A (GABAa) subunits.

Spearman correlation coefficients of mRNA gene expression maps with the GABAa neurotransmitter map ($N$ = 6) (**A**) and their mRNA autocorrelations ($N$ = 6) (**B**). The genes encoding the 19 GABAa subunits include GABRA1–6, GABRB1–3, GABRG1–3, GABRR1–3, GABRD, GABRE, GABRP, and GABRQ. Means are represented by white circles.

findings is that mRNA gene expression seems to vary strongly between individuals. In our mRNA gene expression profile maps, the autocorrelation between mRNA donors was low for 5-HT1b, 5-HT2a, and α1 GABAa subunit, and NET, limiting the confidence in some of these findings. Additionally, the association of mRNA expression with protein products may also vary greatly between genes, being not associated at all or even negatively associated for some, and strongly correlated for others (***Koussounadis et al., 2015***; ***Moritz et al., 2019***). Similarly, a previous study found the correspondence between receptor density and mRNA expression to be low (***Hansen et al., 2022***). Potential reasons

for the lack of or even negative correlations may be a decoupling in time as well as that other levels of regulation overrode the transcriptional level (*Koussounadis et al., 2015*). We observed a similar phenomenon in our data with the correlation of neurotransmitter density maps with their underlying mRNA gene expression being weak for all neurotransmitters except β1 and γ1 GABAa subunits.

Our findings support the notion of fALFF as useful marker for assessing bvFTD-related decline in brain function. In line with previous literature in bvFTD, we observe fALFF reductions mainly in frontal and temporal lobes, but also in the parietal lobe (*Premi et al., 2014*; *Borroni et al., 2018*). These findings support the notion of fALFF being a useful marker of metabolic impairment (*Bang et al., 2015*; *Diehl-Schmid et al., 2007*). Moreover, we found a clear association of fALFF with several neurotransmitter systems pointing to a selective neurotransmitter vulnerability in bvFTD, as suggested in previous research (*Huey et al., 2006*; *Murley and Rowe, 2018*). In particular, the co-localization of fALFF with NET was associated with VF and MMSE, suggesting the sensitivity of fALFF to reflect modality-specific cognitive decline.

The current study was limited by the unavailability of medication information. Therefore, we were not able to control for its potential confounding effects. However, as bvFTD medication is typically restricted to serotonin reuptake inhibitors its effects should be primarily associated with availability of 5-HTT and directionally negate the effects of the disease. Furthermore, as the included PET maps were derived from healthy subjects, the applied approach only tests for co-localization of imaging changes with the non-pathological distribution of the respective neurotransmitter systems. Similarly, the reliability of the co-localization analyses is partly limited by the number of healthy volunteers used to derive the respective neurotransmitter average maps. Finally, the current study was limited by the availability of neurotransmitter maps included in the JuSpace toolbox.

To summarize, we found fALFF reductions in bvFTD to co-localize with the in vivo and ex vivo derived distribution of serotonergic, GABAergic, and norepinephrinergic neurotransmitter systems, pointing to a crucial vulnerability of these neurotransmitters. The strength of these associations was linked to some of the neuropsychological deficits observed in this disease. We propose a combination of spread of pathology through neuronal connectivity and more specifically, through the salience network, as a disease mechanism. Thereby, these findings provide novel insight into the mechanisms underlying the spatial constraints observed in progressive functional and structural alterations in bvFTD. Our data-driven method might even be used to generate new hypotheses for pharmacological intervention in neuropsychiatric diseases beyond this disorder.

## Acknowledgements

This study has been supported by the German Consortium for Frontotemporal Lobar Degeneration, funded by the German Federal Ministry of Education and Research (BMBF; grant no. FKZ01GI1007A). MLS has been furthermore supported by the German Research Foundation (DFG; SCHR 774/5-1) and the eHealthSax Initiative of the Sächsische Aufbaubank (SAB). Accordingly, this study is co-financed with tax revenue based on the budget approved by the Saxon state parliament. JD has received funding from the European Union's Horizon 2020 research and innovation program under grant agreement no. 826421, 'TheVirtualBrain-Cloud'. This work was further supported by the JPND grant 'GENFI-prox' (by DLR/BMBF to MLS, joint with MO). We would like to acknowledge the Clinic for Cognitive Neurology in Leipzig, Annerose Engel, Anke Marschhauser, and Maryna Polyakova.

## Additional information

### Competing interests

Henryk Barthel: received payment or honoraria for lectures, presentations, speakers bureaus, manuscript writing or educational events from Life Molecular Imaging and Novartis/AAA. The author has no other competing interests to declare. Matthis Synofzik: has received consulting fees from, and currently act as a consultant for Aviado Bio, Prevail, Servier, Reata and Orphazyme. They have received payment or honoraria for lectures, presentations, speakers bureaus, manuscript writing or educational events from GenOrph. The author has no other competing interests to declare. Jens Wiltfang: has received consulting fees from Boehringer-Ingelheim, F. Hoffmann-La Roche, Biogen, Immungenetics,

Roboscreen and Abbott. They currently act as a consultant for Boehringer-Ingelheim, F. Hoffmann-La Roche, Biogen and Immungenetics, and hold a Leadership or fiduciary role at CSF Society, AGNP and DGLN. The author has received payment or honoraria for lectures, presentations, speakers bureaus, manuscript writing or educational events from Pfizer, Janssen, MSD SHARP & DOHME, Amgen, Roche Pharma, Actelion Pharmaceutical, Guangzhou Glorylen Medicial Technology Co. (China), Bejing Yibai Science and Technology Ltd. The author has been issued the following patents; EP2095128B1 and EP3105589A1. The author has no other competing interests to declare. Janine Diehl-Schmid: has received a speaker fee from Jansen and Roche. The author has no other competing interests to declare. Markus Otto: has received grants from BMBF - FTLD consortium, moodmarker, ALS association and EU - MIRIADE. The author has received consulting fees from, and currently acts as a consultant for, BIOGEN, Axon and Roche. The author has been issued a patent for Foundation state Baden-Wuerttemberg, Beta Syn as Biomarker for neurodegenerative diseases. The author holds an unpaid leadership or fiduciary role at the German Society for CSF diagnostics and neurochemistry and the Society for CSF diagnostics and neurochemistry, and as a speaker at the FTLD consortium. The author is co-inventor of a patent application (PCT/EP2020/072559) for using beta-synuclein measurement in blood.The author has no other competing interests to declare. Juergen Dukart: former employee of and current consultant for F.Hoffmann-La Roche. The other authors declare that no competing interests exist.

## Funding

| Funder | Grant reference number | Author |
| --- | --- | --- |
| Bundesministerium für Bildung und Forschung | FKZ01GI1007A | Karsten Mueller |
| Deutsche Forschungsgemeinschaft | SCHR 774/5-1 | Matthias L Schroeter |
| Sächsische Aufbaubank | eHealthSax Initiative | Matthias L Schroeter |
| Horizon 2020 - Research and Innovation Framework Programme | TheVirtualBrain-Cloud 826421 | Juergen Dukart |
| EU Joint Programme – Neurodegenerative Disease Research | GENFI-prox | Markus Otto |

The funders had no role in study design, data collection, and interpretation, or the decision to submit the work for publication.

## Author contributions

Lisa Hahn, Formal analysis, Investigation, Visualization, Writing - original draft; Simon B Eickhoff, Conceptualization, Resources, Writing – review and editing; Karsten Mueller, Henryk Barthel, Klaus Fassbender, Klaus Fliessbach, Johannes Kornhuber, Johannes Prudlo, Matthis Synofzik, Jens Wiltfang, Janine Diehl-Schmid, Markus Otto, Matthias L Schroeter, Conceptualization, Data curation, Funding acquisition, Writing – review and editing; Leonhard Schilbach, Conceptualization, Supervision, Writing – review and editing; FTLD Consortium, Conceptualization, Data curation, Funding acquisition; Juergen Dukart, Conceptualization, Supervision, Methodology, Writing – review and editing

## Author ORCIDs

Lisa Hahn http://orcid.org/0000-0003-0167-3947
Simon B Eickhoff http://orcid.org/0000-0001-6363-2759
Klaus Fassbender http://orcid.org/0000-0003-3596-868X
Juergen Dukart https://orcid.org/0000-0003-0492-5644

## Ethics

Written informed consent was collected from each participant. The study was approved by the ethics committees of all universities involved in the German Consortium for Frontotemporal Lobar Degeneration (Ethics Committee University of Ulm approval number 20/10) and was in accordance with the latest version of the Declaration of Helsinki.

Decision letter and Author response
Decision letter https://doi.org/10.7554/eLife.86085.sa1
Author response https://doi.org/10.7554/eLife.86085.sa2

## Additional files

### Supplementary files

• Supplementary file 1. Supplementary tables including information about the subject distribution across centers, detailed information about the neurotransmitter maps, contrast peak voxel information for the t-contrasts, Spearman correlation coefficients and corresponding p-values from the fALFF- and GMV-neurotransmitter analyses and their relationship to clinical symptoms.

• MDAR checklist

### Data availability

The original data and their derivatives cannot be made publicly available as the study includes sensitive patient data and public data sharing was not covered in the informed consent. The original data supporting the findings of this study are available from the senior author (Matthias L. Schroeter) upon reasonable request. All derived statistical measures used here are available from the first author upon request. The software applied is publicly available at https://github.com/juryxy/JuSpace (JuSpace, *Juryxy, 2023*) and https://github.com/FAIR-CNS/MENGA (MENGA, *Rizzo, 2016*). The code for the main analyses is publicly available at https://github.com/liha-coding/Neurotransmitter-vulnerability-in-bvFTD (copy archived at *Hahn, 2023*). Processed data used for the creation of the figures are available as supplementary material.

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
