## [Editor Report]

This study presents important findings linking structural and functional changes in frontotemporal dementia to underlying neurotransmitter systems. The evidence to support the claims is solid, however, relationships are relatively modest. This study will appeal to clinicians and neuroscientists who are interested in the potential effects of certain neurotransmitter systems on clinical features of frontotemporal dementia.

---

## [Decision Letter]

**Decision letter after peer review:**

Thank you for submitting your article "Resting-state alterations in behavioral variant frontotemporal dementia are related to the distribution of monoamine and GABA neurotransmitter systems" for consideration by *eLife*. Your article has been reviewed by 3 peer reviewers, and the evaluation has been overseen by a Reviewing Editor and Timothy Behrens as the Senior Editor. The following individual involved in the review of your submission has agreed to reveal their identity: Vijay Ramanam (Reviewer #3).

Essential revisions:

Introduction:

1. The authors use the raclopride tracer to measure D2 receptor availability in the cortex. According to Dagher and Palomero-Gallagher 2020, this tracer is not sufficiently sensitive to measure the low D2 availability in the cortex. I recommend the authors replace their raclopride tracer for fallypride/FLB-457, or provide justification for using raclopride.

Dagher, A., and Palomero-Gallagher, N. (2020). Mapping dopamine with positron emission tomography: A note of caution. Neuroimage, 207, 116203.

2. The text on "prion-like propagation" in the Introduction seems a bit strong, as this is hardly a settled view in the field (i.e., mechanisms of neurodegenerative disease include many hypotheses, this amongst them).

3. If I understand correctly, "genomic autocorrelation" is identical to gene differential stability (Hawrylycz et al. 2015). I recommend the authors use "differential stability" to keep language consistent in the literature and decrease confusion.

Hawrylycz, M., Miller, J. A., Menon, V., Feng, D., Dolbeare, T., Guillozet-Bongaarts, A. L., … and Lein, E. (2015). Canonical genetic signatures of the adult human brain. Nature neuroscience, 18(12), 1832-1844.

4. The neurotransmitters being considered are themselves a selective list of those involved in mental functioning pathways. Is this a limitation of the toolboxes/data sets utilized, or was this a pre-hoc specification? Would recommend this is discussed in the study limitations.

Methods and Results:

5. Please provide more details of the fMRI metric utilized and the evidence in support of it as a useful biomarker for bvFTD. For example, essentially only one paper describing fALFF as having a correlation with FDG hypometabolism is cited.

6. It would be useful to explain in greater detail how the maps of neurotransmitters are generated with the JuSpace toolbox. I suppose they are based on postmortem human brains. How many does it use, what are the demographics of these brains and how were the neurotransmitters determined?

7. Since the samples at each centre were small, the authors should assess the impact of different parameters in linear models, with fALFF as an output and the different parameters as predictors. Supplementary Table S2 should be moved to the main text so that the readers can see this information as they read the manuscript.

8. What was the rationale behind using 6 mm FWHM smoothing for gray matter volume analyses? This smoothing seems rather low, unless the authors were looking for results in small brain regions, which was not stated. A smoothing of 6 mm for an isotropic fMRI is appropriate but for T1-weighted images is different. Why didn't the authors used DARTEL for their VBM analyses? This greatly improves registration and eliminates potential mismatches between the images.

9. Typical fMRI is highly dependent on the underlying GM structure. Can the authors perform the same fMRI analyses comparing groups while controlling for total GM volume?

10. The major analyses test the fALFF-NT correlation coefficients against neuropsychological test performance. The fALFF-test and NT-test correlations should also be reported, as if either of these are stronger than the main findings it would suggest that there is not a specific fALFF-NT relationship that is really the driver here.

11. For the multiple comparison correction method, why did the authors choose permutation and did not use FDR or FWE provided by SPM? And why did they combine it with an uncorrected threshold? Similarly, the authors state "corrections for autocorrelations were performed" (line 216); could the authors expand on how this was done?

Discussion:

12. Would suggest greater comment in the Discussion on the neuropsychological test analyses. Only a few outcomes showed a nominal relationship with the fALFF-NT coefficients.

13. Sample sizes for D2 and GABAA tracers are quite small (<10) which the authors should explicitly mention in the limitations or replicate with larger-N datasets.

14. The authors motivate studying fALFF in the Introduction but I felt this thread was lost in the Discussion. I recommend the authors expand on (in the Discussion) why fALFF is an appropriate and relevant measure to study in bvFTD, especially in light of their findings.

15. Given the hypothesis that prion-like proteins spread on the underlying structural network, an interesting future direction for this work would be to study structural connectivity instead of GMV. Since the authors mention this hypothesis in the Introduction (line 127), I felt some Discussion on this subject would be of interest, perhaps by expanding the paragraph that starts at line 377?

16. I recommend the authors give credit to previous studies that investigate the mRNA-protein density correlation. Some examples:

a. Beliveau, V., Ganz, M., Feng, L., Ozenne, B., Højgaard, L., Fisher, P. M., … and Knudsen, G. M. (2017). A high-resolution in vivo atlas of the human brain's serotonin system. Journal of Neuroscience, 37(1), 120-128. (For findings regarding the serotonin system)

b. Nørgaard, M., Beliveau, V., Ganz, M., Svarer, C., Pinborg, L. H., Keller, S. H., … and Knudsen, G. M. (2021). A high-resolution in vivo atlas of the human brain's benzodiazepine binding site of GABAA receptors. NeuroImage, 232, 117878. (For findings regarding the GABAA system)

c. Hansen, J. Y., Markello, R. D., Tuominen, L., Nørgaard, M., Kuzmin, E., Palomero-Gallagher, N., … and Misic, B. (2022). Correspondence between gene expression and neurotransmitter receptor and transporter density in the human brain. Neuroimage, 264, 119671. (For showing generally low correspondence between receptor density and mRNA)

d. Rizzo, G., Veronese, M., Heckemann, R. A., Selvaraj, S., Howes, O. D., Hammers, A., … and Bertoldo, A. (2014). The predictive power of brain mRNA mappings for in vivo protein density: a positron emission tomography correlation study. Journal of Cerebral Blood Flow and Metabolism, 34(5), 827. (To my knowledge, the first study to test the association between AHBA mRNA expression and imaging-derived receptor densities across the cortex.)

*Reviewer #1 (Recommendations for the authors):*

(1) The authors use the raclopride tracer to measure D2 receptor availability in the cortex. According to Dagher and Palomero-Gallagher 2020, this tracer is not sufficiently sensitive to measure the low D2 availability in the cortex. I recommend the authors replace their raclopride tracer for fallypride/FLB-45, or provide justification for using raclopride.

Dagher, A., and Palomero-Gallagher, N. (2020). Mapping dopamine with positron emission tomography: A note of caution. Neuroimage, 207, 116203.

(2) Sample sizes for D2 and GABAA tracers are quite small (<10) which the authors should explicitly mention in the limitations or replicate with larger-N datasets.

(3) If I understand correctly, "genomic autocorrelation" is identical to gene differential stability (Hawrylycz et al. 2015). I recommend the authors use "differential stability" to keep language consistent in the literature and decrease confusion.

Hawrylycz, M., Miller, J. A., Menon, V., Feng, D., Dolbeare, T., Guillozet-Bongaarts, A. L., … and Lein, E. (2015). Canonical genetic signatures of the adult human brain. Nature neuroscience, 18(12), 1832-1844.

(4) The authors state "corrections for autocorrelations were performed" (line 216); could the authors expand on how this was done?

(5) I recommend the authors give credit to previous studies that investigate the mRNA-protein density correlation. Some examples:

a. Beliveau, V., Ganz, M., Feng, L., Ozenne, B., Højgaard, L., Fisher, P. M., … and Knudsen, G. M. (2017). A high-resolution in vivo atlas of the human brain's serotonin system. Journal of Neuroscience, 37(1), 120-128. (For findings regarding the serotonin system)

b. Nørgaard, M., Beliveau, V., Ganz, M., Svarer, C., Pinborg, L. H., Keller, S. H., … and Knudsen, G. M. (2021). A high-resolution in vivo atlas of the human brain's benzodiazepine binding site of GABAA receptors. NeuroImage, 232, 117878. (For findings regarding the GABAA system)

c. Hansen, J. Y., Markello, R. D., Tuominen, L., Nørgaard, M., Kuzmin, E., Palomero-Gallagher, N., … and Misic, B. (2022). Correspondence between gene expression and neurotransmitter receptor and transporter density in the human brain. Neuroimage, 264, 119671. (For showing generally low correspondence between receptor density and mRNA)

d. Rizzo, G., Veronese, M., Heckemann, R. A., Selvaraj, S., Howes, O. D., Hammers, A., … and Bertoldo, A. (2014). The predictive power of brain mRNA mappings for in vivo protein density: a positron emission tomography correlation study. Journal of Cerebral Blood Flow and Metabolism, 34(5), 827. (To my knowledge, the first study to test the association between AHBA mRNA expression and imaging-derived receptor densities across the cortex.)

(6) The authors motivate studying fALFF in the Introduction but I felt this thread was lost in the Discussion. I recommend the authors expand on (in the Discussion) why fALFF is an appropriate and relevant measure to study in bvFTD, especially in light of their findings.

(7) Given the hypothesis that prion-like proteins spread on the underlying structural network, an interesting future direction for this work would be to study structural connectivity instead of GMV. Since the authors mention this hypothesis in the Introduction (line 127), I felt some Discussion on this subject would be of interest, perhaps by expanding the paragraph that starts at line 377?

*Reviewer #2 (Recommendations for the authors):*

One of the strengths of the study is the relatively large bvFTD cohort they used, which was recruited from different centres. A limitation associated with this is the fact that the scanning parameters were not exactly the same. For the T1-weighted images they seem relatively harmonised but for fMRI the voxel sizes were not isotropic and display some variability. Since the samples at each centre were small, the authors should assess the impact of different parameters in linear models, with fALFF as an output and the different parameters as predictors.

Supplementary Table S2 should be moved to the main text so that the readers can see this information as they read the manuscript.

What was the rationale behind using 6 mm FWHM smoothing for gray matter volume analyses? This smoothing seems rather low, unless the authors were looking for results in small brain regions, which was not stated.

A smoothing of 6 mm for an isotropic fMRI is appropriate but for T1-weighted images is different. Why didn't the authors use DARTEL for their VBM analyses? This greatly improves registration and eliminates potential mismatches between the images.

Typical fMRI is highly dependent on the underlying GM structure. Can the authors perform the same fMRI analyses comparing groups while controlling for total GM volume?

Which contrast is this: patients>bvFTD? Line 192.

I don't understand the multiple comparison corrections method the authors used. Why did they choose permutation and did not use FDR or FWE provided by SPM? And why did they combine it with an uncorrected threshold?

The reference for the toolbox (JuSpace), which allows to correlate with neurotransmitter systems seems to be an abstract. Has this toolbox been published and used by other groups?

They should recognize the fact that the cognitive results did not survive correction for multiple comparisons as a limitation in the Discussion.

It would be useful to explain in greater detail how the maps of neurotransmitters are generated with the JuSpace toolbox. I suppose they are based on postmortem human brains. How many does it use, what are the demographics of these brains and how were the neurotransmitters determined?

*Reviewer #3 (Recommendations for the authors):*

1. Would suggest a greater explanation of the fMRI metric utilized and the evidence in support of it as a useful biomarker for bvFTD. For example, essentially only one paper describing fALFF as having a correlation with FDG hypometabolism is cited.

2. The neurotransmitters being considered are themselves a selective list of those involved in mental functioning pathways. Is this a limitation of the toolboxes/data sets utilized, or was this a pre-hoc specification? Would recommend this is discussed in the study limitations.

3. The text on "prion-like propagation" in the Introduction seems a bit strong, as this is hardly a settled view in the field (i.e., mechanisms of neurodegenerative disease include many hypotheses, this amongst them).

4. The major analyses test the fALFF-NT correlation coefficients against neuropsychological test performance. The fALFF-test and NT-test correlations should also be reported, as if either of these are stronger than the main findings it would suggest that there is not a specific fALFF-NT relationship that is really the driver here.

5. Would suggest greater comment in the Discussion on the neuropsychological test analyses. Only a few outcomes showed a nominal relationship with the fALFF-NT coefficients.

---

## [Author Response]

Essential revisions:Introduction:1. The authors use the raclopride tracer to measure D2 receptor availability in the cortex. According to Dagher and Palomero-Gallagher 2020, this tracer is not sufficiently sensitive to measure the low D2 availability in the cortex. I recommend the authors replace their raclopride tracer for fallypride/FLB-457 or provide justification for using raclopride.Dagher, A., and Palomero-Gallagher, N. (2020). Mapping dopamine with positron emission tomography: A note of caution. Neuroimage, 207, 116203.

We thank the reviewer for this suggestion and pointing out the weakness of the selected D2 tracer. We have now repeated the analysis using flb457 (please see Table S2 in supplement for the new tracer information). This changed the results so that the fALFF-NT correlation for D2 is no longer significant. We reported the new non-significant results in the main text and removed the respective findings including the follow-up analyses for fALFF-NT correlations with neuropsychological data and fALFF-mRNA correlations.

2. The text on "prion-like propagation" in the Introduction seems a bit strong, as this is hardly a settled view in the field (i.e., mechanisms of neurodegenerative disease include many hypotheses, this amongst them).

We thank the reviewer for this comment. We adjusted the text accordingly to include other theories as well:

“Several possible mechanisms are discussed in the literature for the spread of these proteins throughout the brain, from a selective neuronal vulnerability (i.e., specific neurons being inherently more susceptible to the underlying disease-related mechanisms) to prion-like propagation of the respective proteins.(18,19) The latter entails that misfolded proteins accumulate and induce a self-perpetuating process so that protein aggregates can spread and amplify, leading to gradual dysfunction and eventually death of neurons and glial cells.(19)”

Furthermore, we changed the expression prion-like propagation of the proteins to propagation of the proteins in the other parts of the paper, to be more cautious here.

3. If I understand correctly, "genomic autocorrelation" is identical to gene differential stability (Hawrylycz et al. 2015). I recommend the authors use "differential stability" to keep language consistent in the literature and decrease confusion.Hawrylycz, M., Miller, J. A., Menon, V., Feng, D., Dolbeare, T., Guillozet-Bongaarts, A. L., … and Lein, E. (2015). Canonical genetic signatures of the adult human brain. Nature neuroscience, 18(12), 1832-1844.

We have now changed the term genomic autocorrelation to differential stability in the heading and the main text as well as cited the reference:

“Neurotransmitter-genomic correlations and gene differential stability

To evaluate the robustness of the mRNA maps between donors, gene differential stability was estimated by computing the Fisher’s Z transformed Spearman correlation coefficients between the genomic values of each of the six mRNA donors, which were then averaged.(49)”

4. The neurotransmitters being considered are themselves a selective list of those involved in mental functioning pathways. Is this a limitation of the toolboxes/data sets utilized, or was this a pre-hoc specification? Would recommend this is discussed in the study limitations.

We thank the reviewer for raising this issue. The neurotransmitters were not anyhow preselected but are due to the limited public availability of other PET maps. This is indeed a study limitation and we have now added the following sentence to the discussion:

“Finally, the current study was limited by the availability of neurotransmitter maps included in the JuSpace toolbox.”

Methods and Results:5. Please provide more details of the fMRI metric utilized and the evidence in support of it as a useful biomarker for bvFTD. For example, essentially only one paper describing fALFF as having a correlation with FDG hypometabolism is cited.

We have now provided more details on fALFF. We added a short explanation on the calculation and added a further reference evaluating the correlation of fALFF with FDG PET. Unfortunately, there are indeed only very few publications we are aware of which collected and compared fALFF and FDG-PET measures in the same participants.

fALFF was calculated at each voxel as the root mean square of the blood oxygen level dependent signal amplitude in the analysis frequency band (here: 0.01 – 0.08 Hz) divided by the amplitude in the entire frequency band.(34) fALFF is closely linked to FDG-PET and other measures of local metabolic activity as has been shown in healthy participants but also for example in Alzheimer’s disease.(12,35)

12. Deng S, Franklin CG, O'Boyle M, Zhang W, Heyl BL, Jerabek PA, Lu H, Fox PT. Hemodynamic and metabolic correspondence of resting-state voxel-based physiological metrics in healthy adults. Neuroimage. 2022 Apr 15;250:118923.

35. Marchitelli R, Aiello M, Cachia A, Quarantelli M, Cavaliere C, Postiglione A, Tedeschi G, Montella P, Milan G, Salvatore M, Salvatore E, Baron JC, Pappatà S. Simultaneous resting-state FDG-PET/fMRI in Alzheimer Disease: Relationship between glucose metabolism and intrinsic activity. Neuroimage. 2018 Aug 1;176:246-258. doi: 10.1016/j.neuroimage.2018.04.048. Epub 2018 Apr 27. PMID: 29709628.

6. It would be useful to explain in greater detail how the maps of neurotransmitters are generated with the JuSpace toolbox. I suppose they are based on postmortem human brains. How many does it use, what are the demographics of these brains and how were the neurotransmitters determined?

We thank the reviewer for this comment. The neurotransmitter maps are publicly available PET maps averaged across healthy volunteers hat were acquired in vivo (the details of these maps are provided in Table S3 and also in the original JuSpace publication). To make this more clear, we have now added the following explanations to the Methods section explaining ‘Spatial correlation with neurotransmitter density maps’:

“The JuSpace toolbox allows for cross-modal spatial correlations of different neuroimaging modalities with nuclear imaging derived information about the relative density distribution of various neurotransmitter systems. All neurotransmitter maps were derived as averages from an independent healthy volunteer population and processed as described in the JuSpace publication including rescaling and normalization into the Montreal Neurological Institute space.”

Detailed information about the publicly available neurotransmitter maps is provided in Table S3.

7. Since the samples at each centre were small, the authors should assess the impact of different parameters in linear models, with fALFF as an output and the different parameters as predictors. Supplementary Table S2 should be moved to the main text so that the readers can see this information as they read the manuscript.

We thank the reviewer for this suggestion. We would like to point out that we controlled all analyses for potential site effects to acknowledge the slightly differing acquisition parameters (e.g., by using dummy-coded site variables and regressing out their effect). Furthermore, the literature shows that acquisition parameters such as TR have little to no effect on conventional rsfMRI functional metrics such as fALFF, e.g.:

Huotari N, Raitamaa L, Helakari H, Kananen J, Raatikainen V, Rasila A, Tuovinen T, Kantola J, Borchardt V, Kiviniemi VJ, Korhonen VO. Sampling rate effects on resting state fMRI metrics. Frontiers in Neuroscience. 2019 Apr 2;13:279.

As suggested by the reviewer, we have moved Table S2 to the main text as Table 2.

8. What was the rationale behind using 6 mm FWHM smoothing for gray matter volume analyses? This smoothing seems rather low, unless the authors were looking for results in small brain regions, which was not stated. A smoothing of 6 mm for an isotropic fMRI is appropriate but for T1-weighted images is different. Why didn't the authors used DARTEL for their VBM analyses? This greatly improves registration and eliminates potential mismatches between the images.

We thank the reviewer for this comment. As our motivation for analyzing the structural data was to ensure that the observed functional effects are not driven by the underlying structural changes, we aimed to make the pre-processing as similar as possible between the structural and the resting state data being the main scope of our study. This was also our main reason for using 6mm FWHM for gray matter volume analyses to ensure that a similar smoothing as compared to the derived fALFF maps. For the same reason, we did not use DARTEL as we wanted to keep the normalization parameters identical between both modalities. Moreover, previous research using simulated atrophy showed that the optimal width of smoothing kernel depends on the group size for VBM analyses (Shen and Sterr, 2013). In this publication, for a group size of 50, 6 mm FWHM was recommended as the optimal width of smoothing kernel matching our choice.

Shen S, Sterr A. Is DARTEL‐based voxel‐based morphometry affected by width of smoothing kernel and group size? A study using simulated atrophy. Journal of Magnetic Resonance Imaging. 2013 Jun;37(6):1468-75.

9. Typical fMRI is highly dependent on the underlying GM structure. Can the authors perform the same fMRI analyses comparing groups while controlling for total GM volume?ta

We thank the reviewer for this comment. First, we would like to point out that for this very reason (to make sure that the observed functional effects are not driven by underlying structure), we have initially computed the SPM and JuSpace analysis using GMV as a control analysis to fALFF. In this analysis, GMV and fALFF showed very different results regarding SPM contrasts as well as neurotransmitter co-localization, demonstrating that the observed functional results are clearly distinct from underlying structural changes. Additionally, we have now repeated the SPM contrast and JuSpace analyses controlling for total GMV.

All effects remained significant after controlling for total GMV except for the co-localization with GABAA. We added the results of the SPM contrast analysis as additional row to Table S3 and the results of the JuSpace analysis as additional column (‘fALFF (corrected for GMV)’) to Table S4 in the supplement. Additionally, we briefly describe this analysis in the main text.

To further test if and how the observed fALFF co-localization patterns are explained by the underlying global atrophy, we repeated the co-localization analysis (p<.05) for the significant fALFF-neurotransmitter associations after controlling for total GMV.

When controlling for total GMV, the co-localization findings remained significant except for the co-localization with GABAA.

Cluster size, peak-level MNI coordinates and corresponding anatomical regions incl. the additional fALFF analysis with correction for total GMV are reported in Table S4.

All correlations and their corresponding permutation-based p-values incl. the analysis utilizing fALFF images additionally corrected for total GMV are provided in Table S5.

10. The major analyses test the fALFF-NT correlation coefficients against neuropsychological test performance. The fALFF-test and NT-test correlations should also be reported, as if either of these are stronger than the main findings it would suggest that there is not a specific fALFF-NT relationship that is really the driver here.

We thank the reviewer for this suggestion. We have now examined the fALFF-test correlations using Spearman correlations between the neuropsychological tests and the Eigenvariates of the largest cluster of the controls>patients SPM contrast (please see Table S6 in the supplement). The correlations were not significant after FDR correction. Only the Clinical Dementia Rating for Frontotemporal Lobar Degeneration was significantly correlated with fALFF using an uncorrected threshold of p = 0.05. The weaker nature of these correlations as well as association with different scales both point to the specific nature of the fALFF-NT correlations with neuropsychological tests. Unfortunately, it is not possible to examine the NT-test correlations since the neurotransmitter average maps were collected from different healthy volunteer populations as addressed in point 6 raised by the reviewer. We have now added a description of this analysis and the results to the main text of the manuscript.

Methods:

“In addition, to test for the specificity of these associations we examined the direct associations between fALFF and the neuropsychological tests by computing Spearman correlations with the Eigenvariates extracted from the largest cluster of the HC>bvFTD SPM contrast.”

Results:

“We also tested if Eigenvariates extracted from the largest cluster of the HC>bvFTD contrast correlated with the specific symptoms of bvFTD (Table S6). None of the correlations remained significant after correction for multiple comparisons.”

11. For the multiple comparison correction method, why did the authors choose permutation and did not use FDR or FWE provided by SPM? And why did they combine it with an uncorrected threshold? Similarly, the authors state "corrections for autocorrelations were performed" (line 216); could the authors expand on how this was done?

We thank the reviewer for this comment. Previous research has shown that a family wise error rate of 5% is invalid for clusterwise inference of fMRI data as implemented in SPM and other tools, leading to inflated false-positive rates of up to 70% (Eklund, Nichols, and Knutsson, 2016). The authors propose to use a permutation-based cluster-based threshold together with an uncorrected voxel-based threshold. We have therefore followed this guideline to ensure a proper control for multiple comparisons.

A permutation-based cluster threshold combined with an uncorrected voxel-threshold was used since standard correction methods such as a family wise error rate of 5% may lead to elevated false-positive rates.(37)

37. Eklund A, Nichols TE, Knutsson H. Cluster failure: Why fMRI inferences for spatial extent have inflated false-positive rates. Proceedings of the national academy of sciences. 2016 Jul 12;113(28):7900-5.

The option of adjusting for spatial autocorrelations in the JuSpace toolbox computes partial spatial correlation coefficients by adjusting for local gray matter probabilities as estimated from the TPM.nii provided with SPM12 (Dukart et al., 2020). We clarified this in the respective methods section explaining the JuSpace analysis:

“Furthermore, adjustment for spatial autocorrelation was performed by computing partial correlation coefficients between fALFF and neurotransmitter maps adjusted for local gray matter probabilities estimated from the SPM12-provided TPM.nii.(38)”

38. Dukart J, Holiga S, Rullmann M, Lanzenberger R, Hawkins PCT, Mehta MA, Hesse S, Barthel H, Sabri O, Jech R, Eickhoff SB. JuSpace: A tool for spatial correlation analyses of magnetic resonance imaging data with nuclear imaging derived neurotransmitter maps. Hum Brain Mapp. 2021 Feb 15;42(3):555-566. doi: 10.1002/hbm.25244. Epub 2020 Oct 20.

Discussion:12. Would suggest greater comment in the Discussion on the neuropsychological test analyses. Only a few outcomes showed a nominal relationship with the fALFF-NT coefficients.

We have now expanded the discussion as suggested by the reviewer:

“The strength of co-localization of fALFF with NET was correlated with verbal fluency and MMSE, both being impaired in patients with bvFTD.(4,59,60) Thereby, a stronger negative co-localization (i.e. lower fALFF in patients in high density regions in health) was moderately associated with decreased test performance. Similarly, a correlation between MMSE and NE plasma concentration has been previously reported in Alzheimer’s disease.(61) Combined, these findings point to a potentially more general role of norepinephrinergic neurotransmission in cognitive decline observed across different dementia syndromes. This interpretation is in line with the recently proposed role of the locus coeruleus, the source of norepinephrine in the brain, in regulating processes of learning, memory and attention.(62)”

59. Diehl J, Kurz A (2002): Frontotemporal dementia: patient characteristics, cognition, and behaviour. Int J Geriatr Psychiatry 17: 914–918.

60. Schroeter ML, Pawelke S, Bisenius S, Kynast J, Schuemberg K, Polyakova M, et al. (2018): A modified reading the mind in the eyes test predicts behavioral variant frontotemporal dementia better than executive function tests. Front Aging Neurosci 10: 11.

61. Pillet L-E, Taccola C, Cotoni J, Thiriez H, André K, Verpillot R (2020): Correlation between cognition and plasma noradrenaline level in Alzheimer’s disease: a potential new blood marker of disease evolution. Transl Psychiatry 10: 213.

62. Tsukahara, Jason S., and Randall W. Engle. "Fluid intelligence and the locus coeruleus–norepinephrine system." Proceedings of the National Academy of Sciences 118.46 (2021): e2110630118.

13. Sample sizes for D2 and GABAA tracers are quite small (<10) which the authors should explicitly mention in the limitations or replicate with larger-N datasets.

We thank the reviewer for this suggestion. Based on the first comment listed regarding the tracer for D2, we already repeated the analysis for D2 using a different tracer, resulting in a sample size of 55 subjects. Unfortunately, currently there are no average maps with a larger sample size available for GABAA. Therefore, we added the following limitation to the Discussion section:

“Similarly, the reliability of the co-localization analyses is partly limited by the number of healthy volunteers used to derive the respective neurotransmitter average maps.”

14. The authors motivate studying fALFF in the Introduction but I felt this thread was lost in the Discussion. I recommend the authors expand on (in the Discussion) why fALFF is an appropriate and relevant measure to study in bvFTD, especially in light of their findings.

We agree with the reviewer and have now added a more detailed paragraph about the usefulness of fALFF as an appropriate and relevant measure to study in bvFTD:

“Our findings support the notion of fALFF as useful marker for assessing bvFTD related decline in brain function. In line with previous literature in bvFTD, we observe fALFF reductions mainly in frontal and temporal lobes, but also in the parietal lobe.(13,14) These findings support the notion of fALFF being a useful marker of metabolic impairment.(6,9) Moreover, we found a clear association of fALFF with several neurotransmitter systems pointing to a selective neurotransmitter vulnerability in bvFTD, as suggested in previous research.(24,25) In particular, the co-localization of fALFF with NET was associated with Verbal Fluency and MMSE, suggesting the sensitivity of fALFF to reflect modality specific cognitive decline.”

15. Given the hypothesis that prion-like proteins spread on the underlying structural network, an interesting future direction for this work would be to study structural connectivity instead of GMV. Since the authors mention this hypothesis in the Introduction (line 127), I felt some Discussion on this subject would be of interest, perhaps by expanding the paragraph that starts at line 377?

We agree with this comment and therefore added the following sentences in the discussion:

“An interesting future direction might compose integration of structural connectivity as measured by diffusion tensor imaging. A study by Dopper and colleagues (2014) showed reduced fractional anisotropy in healthy individuals carrying mutations compared to non-carriers.(58) Given that there were structural connectivity differences even before disease onset, it would be of interest to re-examine structural connectivity differences between healthy controls and patients (i.e. after disease onset). Repeating the neurotransmitter analyses might facilitate understanding of the underlying disease mechanism.”

58. Dopper EGP, Rombouts SARB, Jiskoot LC, Heijer T den, Graaf JRA de, Koning I de, et al. (2014): Structural and functional brain connectivity in presymptomatic familial frontotemporal dementia. Neurology, vol. 83. pp e19–e26.

16. I recommend the authors give credit to previous studies that investigate the mRNA-protein density correlation. Some examples:a. Beliveau, V., Ganz, M., Feng, L., Ozenne, B., Højgaard, L., Fisher, P. M., … and Knudsen, G. M. (2017). A high-resolution in vivo atlas of the human brain's serotonin system. Journal of Neuroscience, 37(1), 120-128. (For findings regarding the serotonin system)b. Nørgaard, M., Beliveau, V., Ganz, M., Svarer, C., Pinborg, L. H., Keller, S. H., … and Knudsen, G. M. (2021). A high-resolution in vivo atlas of the human brain's benzodiazepine binding site of GABAA receptors. NeuroImage, 232, 117878. (For findings regarding the GABAA system)c. Hansen, J. Y., Markello, R. D., Tuominen, L., Nørgaard, M., Kuzmin, E., Palomero-Gallagher, N., … and Misic, B. (2022). Correspondence between gene expression and neurotransmitter receptor and transporter density in the human brain. Neuroimage, 264, 119671. (For showing generally low correspondence between receptor density and mRNA)d. Rizzo, G., Veronese, M., Heckemann, R. A., Selvaraj, S., Howes, O. D., Hammers, A., … and Bertoldo, A. (2014). The predictive power of brain mRNA mappings for in vivo protein density: a positron emission tomography correlation study. Journal of Cerebral Blood Flow and Metabolism, 34(5), 827. (To my knowledge, the first study to test the association between AHBA mRNA expression and imaging-derived receptor densities across the cortex.)

We thank the reviewer for this comment. We have now cited the sources of all neurotransmitter maps utilized in our study within the methods. More detailed information can be found in Table S2 in the supplement. We additionally cited the Rizzo reference together with their MENGA toolbox in the methods and the Hansen reference in the discussion:

“Similarly, a previous study found the correspondence between receptor density and mRNA expression to be low.(65)”